# Measurement of Anti-TNF Biologics in Serum Samples of Pediatric Patients: Comparison of Enzyme-Linked Immunosorbent Assay (ELISA) with a Rapid and Automated Fluorescence-Based Lateral Flow Immunoassay

**DOI:** 10.3390/pharmaceutics17040421

**Published:** 2025-03-26

**Authors:** Chiara Rossi, Raffaele Simeoli, Giulia Angelino, Sara Cairoli, Fiammetta Bracci, Daniela Knafelz, Erminia Francesca Romeo, Simona Faraci, Giusyda Tarantino, Alessandro Mancini, Alessia Vitale, Carlo Dionisi Vici, Silvia Magni Manzoni, Paola De Angelis, Bianca Maria Goffredo

**Affiliations:** 1Division of Metabolic Diseases and Hepatology, Bambino Gesù Children’s Hospital, IRCCS, 00165 Rome, Italy; chiara.rossi@opbg.net (C.R.); sara.cairoli@opbg.net (S.C.); alessandro.mancini@opbg.net (A.M.); alessia.vitale@opbg.net (A.V.); carlo.dionisivici@opbg.net (C.D.V.); 2Digestive Endoscopy and Surgery Unit, Bambino Gesù Children’s Hospital, IRCCS, 00165 Rome, Italy; giulia.angelino@opbg.net (G.A.); efrancesca.romeo@opbg.net (E.F.R.); paola.deangelis@opbg.net (P.D.A.); 3Hepatology and Gastroenterology Unit, Bambino Gesù Children’s Hospital, IRCCS, 00165 Rome, Italy; fiammetta.bracci@opbg.net (F.B.); daniela.knafelz@opbg.net (D.K.); simona.faraci@opbg.net (S.F.); 4Rheumatology Department, Bambino Gesù Children’s Hospital, IRCCS, 00165 Rome, Italy; giusyda.tarantino@opbg.net (G.T.); silvia.magnimanzoni@opbg.net (S.M.M.)

**Keywords:** pediatric patients, therapeutic drug monitoring (TDM), anti-TNF biological agents, enzyme-linked immunosorbent assay (ELISA), fluorescence-based lateral flow immunoassay (AFIAS), turnaround time (TAT)

## Abstract

**Background**: Therapeutic drug monitoring (TDM) of infliximab (IFX) and adalimumab (ADL) mainly relies on the use of enzyme-linked immunosorbent assays (ELISA). More recently, rapid assays have been developed and validated to reduce turnaround time (TAT). Here, we compared IFX and ADL concentrations measured with both ELISA and a new fluorescence-based lateral flow immunoassay (AFIAS). **Methods**: In serum samples from pediatric patients, IFX and ADL drug levels, and total anti-IFX antibodies were measured using clinically validated ELISA kits (Immundiagnostik AG). Samples were further analyzed using a new rapid assay (AFIAS, Boditech Med Inc.) to measure drug levels and total anti-IFX antibodies. **Results**: Spearman’s correlation coefficients (rho) were 0.98 [95% confidence interval (CI) 0.97 to 0.99] for IFX (*p* < 0.001) and 0.83 (95% CI 0.72 to 0.90) for ADL (*p* < 0.001). Calculated % bias was −14.09 (95% Limits of agreement, LoA, −52.83 to 24.66) for IFX and 15.79 (LoA −37.14 to 68.73) for ADL. For the evaluation of total anti-IFX antibodies, we did not collect sufficient data to establish a statistically significant correlation between AFIAS and ELISA. The inter-rater agreement showed a “substantial” and a “moderate” agreement for IFX and ADL, respectively. **Conclusions**: Our results show that the AFIAS assay has an accuracy and analytical performance comparable to that of the ELISA method used for TDM of IFX and ADL. Therefore, the introduction of this device into routine clinical practice could provide results more quickly and with similar accuracy as ELISA, allowing clinicians to rapidly formulate clinical decisions.

## 1. Introduction

Inflammatory bowel diseases (IBDs), including Crohn’s disease (CD) and ulcerative colitis (UC), are extremely debilitating conditions that can significantly affect patients’ quality of life. Inflammatory bowel diseases affect approximately 25% of pediatric patients, and often presents with a more serious disease course compared with adult IBDs [1].

However, owing to significant progress made in biotechnological sciences, medical treatment of IBD has been greatly improved by the introduction of biological agents capable of targeting pro-inflammatory cytokines, such as tumor necrosis factor (TNF)-α, interleukin (IL)-12/23, and α_4_β_7_ integrin, which are involved in immune responses [2].

Monoclonal antibodies (mAb) directed against TNF-α (i.e., infliximab, IFX, and adalimumab, ADL) were the first biologics introduced more than a decade ago as therapeutic options for IBD and are still widely used for the treatment of moderate to severe forms of disease [3,4]. In particular, according to the ECCO-ESPGHAN Guideline Update for the Medical Management of Pediatric Crohn’s Disease, in new-onset patients at high risk for a complicated disease course, anti-TNF therapy is recommended for inducing remission [5]. Similarly, anti-TNF agents are recommended for induction and maintenance of remission in patients with active CD who do not achieve or maintain remission [5].

Moreover, the introduction of these agents has significantly improved the quality of life of patients, thereby reducing the need for surgical approaches [6].

However, despite their beneficial therapeutic effects, following a primary response to anti-TNF therapy, patients treated with these agents may experience loss of response (LoR) and relapse of the disease [7]. The reasons for these therapeutic failures are not yet fully understood; however, inter-individual variability in drug pharmacokinetics (PK), pharmacodynamics (PD), and immunogenicity are considered possible risk factors [8]. The risk of developing anti-drug antibodies (ADAs) is a serious complication of biological therapies and is often associated with poor response to these treatments [9]. There are multiple mechanisms underlying the development of immunogenicity; however, the presence of specific pharmacogenetic variants and/or prolonged sub-therapeutic drug levels seems to be positively correlated [10,11,12]. It is also worthwhile to say that a successful therapy with anti-TNF biological agents relays not only on the immunogenicity development but also on the choice of an adequate dosing regimen during both the induction and maintenance of remission [13]. In fact, patients classified as non-responders often display sub-therapeutic drug levels, although typical infusion regimens are given [14]. In addition, considering that biological therapies can also present several side effects and are characterized by high costs, the optimization of tailored therapies should be a desirable endpoint during routine clinical practice.

In this regard, therapeutic drug monitoring (TDM) can be used not only to assess whether low drug levels (with or without the presence of ADAs) could be responsible for therapeutic failures (reactive TDM), but also to allow dose adjustments and ensure that drug levels are within a therapeutic window, thus avoiding risks of future disease relapses (proactive TDM) [15]. In fact, TDM of anti-TNF drug levels and anti-drug antibodies allows for the identification of patients who have lost response to the biological agent but also to potentially define a tailored therapeutic strategy [16]. Specifically, TDM assesses the adequacy of drug plasma concentrations in relation to a target concentration or concentration window at a specific time within a dosing interval [17]. Previously, it has been reported that both primary non-response and secondary LOR in anti-TNF-treated patients commonly result from low trough concentration, high ADA levels, or both [18,19,20]. Based on this evidence, the ECCO-ESPGHAN Guideline 2020 states that early proactive therapeutic drug monitoring followed by dose optimization is recommended [5].

Proactive TDM is defined as the evaluation of trough concentrations and ADA levels, with the aim of optimizing biological therapy to achieve an effective threshold drug concentration [15]. Moreover, recent data have reported that proactive TDM is associated with better therapeutic outcomes than empirical dose optimization and/or reactive TDM of anti-TNF agents in IBD [21,22]. To date, several TDM-based algorithms have been proposed in real-life clinical practice to guide the tailored optimization of anti-TNF therapies [15,23]. Similarly, clinical outcome predictions in pediatric IBD patients have been suggested by measuring infliximab trough levels at different sampling points during the induction and maintenance phase [15,18].

In fact, proactive TDM allows the maintenance of relatively constant drug levels, creating a personalized and effective therapy and identifying patients suffering from inadequate exposure. Monitoring the serum concentration of the administered drug and the concentration of anti-drug antibodies (ADA) is a fundamental aspect that must be monitored during all phases of therapy, starting from the induction and maintenance phases to complete remission. Only in this way will it be possible to guarantee adequate exposure and reduce the risk of loss of response.

Therefore, based on the evidence that measuring anti-TNF agents and ADA levels improves patient outcomes and could potentially be more cost-effective, several bioanalytical assays have been developed to test both drug and ADA concentrations in serum and plasma samples [16]. These include drug-sensitive solid-phase immunoassays, radioimmunoassay (RIA), homogeneous mobility shift assays (HMSA), and cell reporter gene assays (RGA) [18]. Recently, faster assays, such as fiber-optic surface plasmon resonance (FO-SPR) and lateral flow (LF) tests, have been introduced in the market [22]. Enzyme-linked immunosorbent assay (ELISA) is the most commonly used immunoassay to detect both drug and ADA levels in different biological matrices [24]. Despite its wide adoption across bioanalytical laboratories to measure different analytes, ELISA presents several disadvantages, including time-consuming incubation and washing steps and the need to obtain a certain number of samples before proceeding with evaluation (test in batches). Because ELISA assays require the construction of a calibration curve each time, filling a whole plate by accumulating the desired number of samples could be a valid strategy to reduce costs. Taken together, these aspects can significantly increase the turnaround time (TAT) required to obtain the TDM results. Therefore, in the last decade, there has been growing attention on point-of-care (POC) assays that allow the measurement of drug and anti-drug levels at patient’s bedside or in an outpatient clinic in a shorter time (15–20 min) compared to ELISA (3–5 h) [25]. In a study conducted by Curci D and colleagues (2019), two different POC devices based on lateral flow assays were validated to measure serum infliximab levels by comparing their results with those obtained using a reference ELISA kit [26]. More recently, a similar approach was adopted to evaluate the analytical and clinical performance of a new fluorescence-based lateral flow immunoassay (AFIAS) by measuring and comparing infliximab and adalimumab drug levels with those obtained through commercially available ELISA kits [14,27,28]. This rapid assay (20 min) allows the quantitative measurement of IFX, ADL, and anti-drug antibodies on a cartridge ready for direct use on a fully automated instrument (AFIAS). Moreover, the samples did not require repeated dilution, mixing, or separation steps [14,27,28].

So far, previous studies have already compared infliximab drug levels measured with both AFIAS and the gold-standard ELISA in samples from pediatric and adult patients [14,27,28]. However, to the best of our knowledge, this is the first study in which, alongside with AFIAS IFX, the analytical performance of AFIAS ADL has also been evaluated in serum samples collected from pediatric patients during the routine clinical practice. To this aim, here we have used the AFIAS system to quantify IFX and ADL levels in serum samples from pediatric patients treated with these biological agents. Anti-IFX antibody levels were also measured in the same samples. AFIAS results were then compared with those obtained by using a commercially available ELISA kit considered as a reference assay in our routine clinical practice.

## 2. Materials and Methods

### 2.1. Patients’ Samples

Samples were obtained from 98 hospitalized pediatric and/or outpatient patients who received biological therapy with infliximab (IFX) or adalimumab (ADL). In particular, n = 44 serum samples were collected for therapeutic drug monitoring (TDM) of IFX and anti-IFX levels, and n = 54 samples were obtained to monitor ADL serum levels. The mean ± SD age of the pediatric patients was 15 ± 9.55 for the IFX group and 15 ± 4.45 ADL. Blood samples were collected immediately before the next dose was administered; therefore, the trough levels (C_trough_) were monitored for both infliximab and adalimumab. Following centrifugation at 3.500 x g for 10 min, serum samples were collected, and one aliquot (approximately 300 μL) was used for AFIAS tests, while another aliquot was stored at −80 °C until ELISA assays were conducted. Both tests were performed at the Laboratory of Metabolic Diseases and Hepatology of Bambino Gesù Children’s Hospital, IRCCS, in Rome (Italy).

TDM of Infliximab and Adalimumab is well established in the routine clinical practice at Bambino Gesù Children’s Hospital. This study was conducted in full observation of the Helsinki Declaration and the protocol was approved by the Ethics Committee of Bambino Gesù Children’s Hospital (2333_OPBG_2020) on 4 January 2021.

### 2.2. Enzyme-Linked Immunoassorbent Assays (ELISA)

Infliximab or Adalimumab levels were determined using commercially available and validated ELISA kits (IDKmonitor^®^, Immunodiagnostik AG, Bensheim, Germany) according to the manufacturer’s instructions. These ELISA assays are designed to quantitatively measure free infliximab or adalimumab levels in serum and EDTA plasma samples, and involve several steps. Briefly, during the first incubation, the free drug contained in the unknown samples was bound to specific anti-drug antibodies coated on a 96-well plate. A washing step was performed to remove all the unbound substances. Finally, after another incubation step, in which a peroxidase-labeled secondary antibody was added to each well, the use of a peroxidase substrate (tetramethylbenzidine-TMB) determined the progress of the reaction, resulting in the formation of a blue-colored product. An acid-containing stop solution was added to terminate the reaction and allow color variation from blue to yellow. The intensity of the yellow color is directly proportional to the concentration of the free drug present in the sample. Using a spectrophotometer, the plate was read at 450 and 620 nm (reference wavelength), and optical densities (ODs) were used to construct a 4PL sigmoidal curve using the calibration standards provided by the kit. The drugs’ concentrations in each sample were determined directly by fitting the optical density (OD) values to this curve. Finally, IFX or ADL levels were calculated, multiplying the obtained results by a dilution factor of 200. Results were expressed as µg/mL.

For each ELISA kit (Cat n. K9655 IDKmonitor^®^ infliximab drug level and n. K9657 IDKmonitor^®^ adalimumab drug level), Quality Controls at low (L-QC) and high (H-QC) drug concentrations with relative target ranges were provided by the manufacturer. For infliximab drug levels, target ranges were 7.5–19.5 ng/mL and 39.0–89.0 ng/mL for L-QC and H-QC, respectively. For adalimumab drug levels, target ranges were as follows: 0.9–10.9 ng/mL and 20.0–70.0 ng/mL for L-QC and H-QC, respectively. The validity of the tests was evaluated by analyzing the QCs together with the unknown samples in each analytical batch.

Similarly, the determination of infliximab total ADA was performed on the same IFX serum samples using an ELISA kit (Cat n. K9654 IDKmonitor^®^ Infliximab Total ADA) provided by Immunodiagnostik AG (Bensheim, Germany). This validated ELISA allows for the semi-quantitative detection of free and bound (total) anti-drug antibodies, even in the presence of infliximab. According to the manufacturer’s instructions, serum samples were subjected to preliminary incubation that allowed the separation anti-drug antibodies (ADA) from the therapeutic antibody to detect free ADA. After this incubation step, by adding a conjugate (peroxidase-labeled therapeutic antibody) and tracer (biotinylated therapeutic antibody) to each sample, the unmarked therapeutic antibodies were replaced, and the marked antibodies formed a complex with ADA. This complex binds via biotin to streptavidin-coated microtiter plates via biotin. Thereafter, the complex was detected using a peroxidase conjugate that converted the TMB substrate into a blue product. Finally, the enzymatic reaction was stopped by adding an acid solution, which changed the color of the samples from blue to yellow. The color change was measured using a spectrophotometer at 450 and 620 nm (reference wavelengths). The results were interpreted using the cut-off control sample according to the manufacturer’s instructions provided by Immunodiagnostik AG. (Bensheim, Germany). The validity of the tests was assessed by analyzing a Positive and a Negative Control sample provided by the kit, together with unknown samples in each analytical batch. The analytical specifications of the ELISA kits used in this study are listed in Appendix A.

### 2.3. Fluorescence-Based Lateral Flow Immunoassay

Infliximab and adalimumab levels were measured in the same serum samples alongside with ELISA by using commercially available kits on the AFIAS-6 instrument (Bodytech Med Inc., 43, Geodudanji 1-gil, Dongnae-myeon, Chuncheon-si, Gangwon-do, Republic of Korea). This fluorescence-based lateral flow immunoassay was developed and validated for the quantitative analysis of IFX (Cat. n. SMFP-75 AFIAS Infliximab) and ADL (SMFP-89 AFIAS Adalimumab) drug levels in whole blood, plasma, and serum samples (Bodytech Med Inc., Republic of Korea). Moreover, AFIAS allows the semi-quantitative analysis of total antibodies against infliximab in both plasma and serum (Cat. n. SMFP-76 AFIAS Total Anti-Infliximab, Bodytech Med Inc., Republic of Korea). AFIAS IFX and ADL were prepared as all-in-one cartridges for direct use and applied to a fully automated instrument, AFIAS (Bodytech Med Inc., Republic of Korea).

For cartridge preparation, three components were used with the kit: a Sample Dilution Buffer, conjugated antibodies, and a nitrocellulose strip.

IFX and ADL levels were measured according to the manufacturer’s instructions. Briefly, the assay procedure involved loading the sample (100 µL) into the sample into a specific well on the cartridge. Thereafter, the cartridge was inserted into the cartridge holder in the AFIAS instrument, where the samples were automatically diluted and mixed with the AFIAS reagent. The fluorescence and biotin-labeled anti-infliximab in the buffer bound to infliximab or adalimumab was present in each sample, forming drug–antibody complexes, and migrating onto the nitrocellulose matrix to be captured by the other immobilized streptavidin on a test strip. More drugs are present in the samples, and more drug–antibody complexes will form, leading to a stronger fluorescence signal. Finally, the test results were automatically calculated from a previously loaded calibration ID chip (lot specific) and displayed on the screen within 10 min.

Quality Controls for low (L-QC) and high (H-QC) drug concentrations with relative target ranges were provided by the manufacturer. For infliximab drug levels, target ranges were 1.5–2.5 μg/mL and 15.0–25.0 μg/mL for L-QC and H-QC, respectively. For adalimumab drug levels, ranges were 2.7–6.3 μg/mL and 11.4–26.6 μg/mL, respectively.

The validity of the tests was evaluated by analyzing the QCs together with the unknown samples before each analytical session.

Similarly, total antibodies (free and bound forms) directed against infliximab were semi-quantitatively measured on the same IFX serum samples using a validated AFIAS Total Anti-Infliximab kit (Bodytech Med Inc., Republic of Korea). This test uses a bridging immunoassay to detect antibodies against infliximab. The assay is based on a system similar to that adopted for the detection of drug levels. Briefly, after loading on the cartridge, the sample was subjected to dissociation, and anti-drug antibodies from the samples were bridged with fluorescence and biotin-labeled infliximab, which resulted in the formation of an immune complex. This immune complex migrates into the nitrocellulose matrix and is captured by immobilized streptavidin on a test strip. More antibodies to infliximab are present in the sample, and more immune complexes will form, leading to stronger intensity of the fluorescence signal, which is processed by the instrument to show total anti-infliximab concentration in a working range of 8–250 AU/mL.

The analytical specifications of the AFIAS kits used in this study are listed in Appendix A.

### 2.4. Stability

Since ELISA test was performed on frozen and thawed samples, we have evaluated long-term stability by re-assessing infliximab and adalimumab concentrations on randomly selected authentic samples (n = 4) that were stored at −80 °C over a period of one month. The ratio of the concentration after a second freezing/thawing cycle to the initial measurement was used to calculate a percentage difference. Based on EMA guidelines, an acceptable stability should have a percentage difference below 15%.

### 2.5. Statistical Methods

Nonparametric Passing–Bablok regression and Spearman correlation coefficients were used to evaluate the agreement between drug concentrations measured with both ELISA and AFIAS assays. For the slope and intercept, 95% confidence intervals (CIs) were calculated. Thereafter, the Bland–Altman test was applied to assess the relative differences between the two assays by plotting the % differences against the average anti-TNF agent concentrations measured in serum using ELISA and AFIAS. The mean difference (bias) and standard deviation (SDs) of the mean were calculated. According to The ICH guideline M10 on bioanalytical method validation and study sample analysis. 25 July 2022, EMA/CHMP/ICH/172948/2019 Committee for Medicinal Products for Human Use. Available at https://www.ema.europa.eu/en/ich-m10-bioanalytical-method-validation-scientific-guideline, accessed on 28 August 2023), the difference between the two values obtained should be within 20% of the mean for at least 67% of the repeats. For a qualitative comparison between IFX and ADL drug levels measured by both ELISA and AFIAS, weighted kappa statistics were determined after the stratification of results according to the therapeutic range (IFX < 3 µg/mL, ≥3 to <7 µg/mL, and ≥7 µg/mL) (ADL < 5 µg/mL, ≥5 to <10 µg/mL, and ≥10 µg/mL). Therapeutic intervals were determined according to previously published reports [26,29,30]. Drugs levels measured with both ELISA and AFIAS were expressed as median and interquartile range (IQR). Mann–Whitney test was used to compare two groups of non-parametric data. All statistical analyses were performed using GraphPad Prism, version 10 (GraphPad Software, San Diego, CA 92108, USA).

## 3. Results

### 3.1. Comparison of IFX and ADL Serum Levels Measured with Both AFIAS and ELISA Assays

The median concentration of IFX serum levels measured with AFIAS was 4.38 µg/mL (IQR: 5.57), while the median concentration of ELISA measurements was 4.92 µg/mL (IQR: 5.90). A significant difference was not observed in IFX trough levels determined with both assays (*p* = 0.445). Similarly, the median of ADL trough concentrations measured with AFIAS was 8.73 µg/mL (IQR: 6.07), whereas the median of ADL values found in the same serum samples by using ELISA was 7.09 (IQR: 3.94). Among AFIAS and ELISA ADL trough levels, we have not observed a significant difference (*p* = 0.135). Paired raw data of IFX and ADL concentrations measured with both methods are reported in Appendix A. Figure 1 shows the Passing–Bablok correlation plot for IFX and ADL serum concentrations measured by AFIAS and ELISA. For IFX levels, the fit was y = 1.25X − 0.28 (95% CI slope, 0.83 to 1.67; 95% CI intercept, −2.11 to 1.55) (Figure 1A), meanwhile for ADL it was y = 0.72X + 1.07 (95% CI slope, 0.58 to 0.86; 95% CI intercept, −0.04 to 2.19) (Figure 1B). For IFX and ADL, there was an absence of no constant bias, as both the 95% CI intercepts included 0. Exclusively for ADL, the 95% CI slopes did not include 1, indicating the presence of a slight proportional bias.

The Spearman’s correlation coefficients (rho) were 0.98 (95% CI 0.97 to 0.99) for IFX (Figure 1A) and 0.83 (95% CI 0.72 to 0.90) for ADL (Figure 1B), indicating a positive correlation between drug concentrations measured with both assays. Moreover, a significant *p* value was observed for both IFX (n = 44) and ADL (n = 54) (*p* < 0.001).

The results were further evaluated using the Bland–Altman test (Figure 1). In these graphs, the data are displayed as scatter diagrams of the % difference plotted against the average of two measurements. Continuous horizontal blue lines were drawn at the mean difference (bias) and 95% limits of agreement, which are defined as the mean difference ± 1.96 times the standard deviation of the differences. Dotted horizontal black lines indicate the 95% confidence intervals (CI) for the mean and the agreement limits. The % mean difference values were −14.09 (95% CI −8.08 to −20.1) and 15.79 (95% CI 8.42 to 23.13) for IFX and ADL, respectively (Figure 1C,D).

### 3.2. Comparison of Total Anti-Infliximab Concentrations in Serum Samples Obtained by Using AFIAS and ELISA Assays

AFIAS Total Anti-infliximab was used for the semi-quantitative detection of antibodies directed against infliximab. For this assay, a working range of 8–250 AU/mL was used, and values below or above this range were displayed as <8 and >250 AU/mL, respectively. Therefore, we did not collect sufficient data to establish a statistically significant correlation between AFIAS and ELISA. However, 39 serum samples were analyzed using AFIAS, which showed a concentration of <8 AU/mL. These samples were also tested using an ELISA kit, which resulted in a median (range) value of 4.78 (2.22–12.31) AU/mL. Similarly, among the AFIAS results, n = 4 serum samples presented a total antibody concentration >250 AU/mL. The median value obtained with the ELISA assay was 576.5 AU/mL (range 363.1–771.8 AU/mL).

### 3.3. Qualitative Evaluation of Agreement Between Drug Levels Measured with Both AFIAS and ELISA Assays

To qualitatively compare the serum drug levels obtained with the AFIAS system and the ELISA assay, we evaluated the inter-rater agreement with kappa by stratifying the results into three subgroups according to the therapeutic interval. For IFX, we divided the measurements as follows (<3 µg/mL, ≥3 to ≤7 µg/mL, and ≥7 µg/mL), based on a previously described therapeutic range of 5–7 µg/mL [26,29] (Table 1). Infliximab levels showed “substantial agreement” between the two assays according to the scale suggested by Landis and Koch, G.G. [31]. Overall, the weighted kappa was 0.73 (95% CI 0.58, 0.87), and the number of observed agreements between the reference ELISA and the AFIAS assay was 33/44 (75% of the observations). In particular, for the n = 13 samples with IFX levels <3 µg/mL (as measured with reference ELISA assay), an agreement was achieved for 13/13 (100%) samples tested with AFIAS; for samples with infliximab drug levels between 3 and 7 µg/mL (n = 13) obtained with the reference ELISA, agreement was reached for 8/13 (61.5%) of AFIAS samples; and finally, when IFX levels for the reference ELISA were > 7 µg/mL (n = 18 samples), 12/18 (66.6%) of AFIAS measurements were in agreement with it. Similarly, ADL drug levels were stratified into three subgroups (<5 µg/mL, ≥5 to ≤10 µg/mL, and ≥ 10 µg/mL) according to the therapeutic range of 5–10 µg/mL proposed for inflammatory bowel diseases [30] (Table 2). The weighted kappa for adalimumab was 0.52 (95% CI 0.34 to 0.71) denoting a “moderate agreement” [31]. For ADL, the overall number of observed agreements between reference ELISA and AFIAS assay was 35/54 (64.8% of the observations). In detail, for the n = 12 ADL samples with drug levels < 5 µg/mL (measured by ELISA as reference method), agreement was reached for 8/12 (66.6%) samples tested with AFIAS as well; for samples with adalimumab drug levels between 5 and 10 µg/mL (n = 29) measured by ELISA, agreement was obtained for 18/29 (62%) of AFIAS measurements; and finally, at drug levels > 10 µg/mL for the reference ELISA (n = 13), 9/13 (69.2%) samples were in agreement with the results of AFIAS.

Conversely, when we have stratified IFX drug levels measured with AFIAS, at infliximab levels <3 µg/mL (n = 18), agreement with ELISA was achieved in 13/18 samples (72.2%); for AFIAS results comprised between 3 and 7 µg/mL (n = 14), agreement with ELISA was reached in 8/14 (57.1%) measurements; and finally, when AFIAS drug levels were >7 µg/mL (n = 12), all the measurements (12/12, 100%) were in agreement with the ELISA results. Similarly, by stratifying AFIAS results for ADL drug levels based on the therapeutic range 5–10 µg/mL, at adalimumab concentrations < 5 µg/mL (n = 10), the agreement with ELISA was 8/10 (80%). For AFIAS results between 5 and 10 µg/mL (n = 26), agreement with ELISA was found in 18/26 (69.2%) measurements; and finally, at AFIAS drug levels >10 µg/mL (n = 18), 9/18 (50%) samples agreed with the ELISA assay.

### 3.4. Evaluation of Stability

In order to exclude a possible bias due to the freezing–thawing procedure applied to the ELISA samples, we have randomly selected and re-analyzed n = 4 authentic samples after being stored at −80 °C over a period of one month. Therefore, a second freezing–thawing cycle was executed 30 days after the first one, and revealed a stability of 119.0 ± 15.26% (mean ± SD) for IFX and 97.52 ± 14.98% for ADL compared to the initial drugs’ measurements.

## 4. Discussion

According to the ECCO-ESPGHAN Guideline 2020, early proactive therapeutic drug monitoring followed by dose optimization is recommended for patients treated with anti-TNF agents [5]. Proactive TDM is defined as the evaluation of trough concentrations and ADA levels, with the aim of optimizing biological therapy to achieve an effective threshold drug concentration [15]. In contrast to reactive TDM, which is defined as the evaluation of drug concentration and ADA levels in case of primary non-response or secondary loss of response (LoR) to a biological agent, preliminary data suggest that proactive TDM of anti-TNF therapy is associated with better therapeutic outcomes than cost-expansive empirical dose optimization and/or reactive TDM [15]. Although the introduction of TNF biological blockers into clinical practice has significantly improved the treatment of inflammatory conditions such as IBDs and juvenile idiopathic arthritis, patients treated with these agents may experience LoR and relapse of the disease [7]. The reasons for these therapeutic failures are not yet fully understood; however, inter-individual variability in drug pharmacokinetics (PK), pharmacodynamics (PD), and immunogenicity are considered possible risk factors [8,15,21]. Although different bioanalytical methods have been proposed to measure anti-TNF agents in both serum and plasma samples, ELISA remains the most commonly used assay [18]. However, over the last few decades, point-of-care (POC) devices based on lateral flow (LF) immunoassays have been developed and validated for measuring both drug and ADA levels with a rapid turnaround time [14,26,27,28]. Despite ELISA being a reliable technology characterized by the possibility of automating tests, lower costs, and high-throughput analyses, it also presents disadvantages. Among them, ELISA requires several hours of work (4–8 h) executed by appropriately trained personnel, and it is cost-effective only if patient samples are accumulated and tested in batches [18]. Consequently, TAT is significantly prolonged and TDM results are not immediately available to clinicians. Conversely, POC devices allow for single-patient analyses and provide both quantitative and semiquantitative results within 15–20 min [14,27,28,32,33,34].

Nevertheless, before being adopted by a biochemistry laboratory working on the TDM of anti-TNF blockers, these POC devices should be adequately cross-validated using different bioanalytical assays. In this regard, several reports have been published to date comparing ELISA with different POC tests for the measurement of both drug and ADA levels in serum or plasma samples [14,26,27,28,32,33,34,35,36,37]. In our study, we have used the AFIAS system (Bodytech Med Inc., Republic of Korea) to measure IFX and ADL drug levels in serum samples collected from pediatric patients during the routine clinical practice. Additionally, anti-IFX Total ADAs were also tested in the same IFX serum samples.

Both AFIAS IFX and AFIAS ADL are fluorescence-based lateral flow immunoassays that have been successfully developed and validated for quantitative determination of infliximab and adalimumab in different biological matrices, including whole blood, plasma, and serum (Bodytech Med Inc., Republic of Korea). To date, the analytical and clinical performance of these assays have been already evaluated by comparing drugs’ concentrations measured with both AFIAS and different ELISA kits used for TDM analysis in adult and pediatric patients [14,27,28]. Results were comparable among the different analytical methods, showing a good correlation between AFIAS and the reference ELISA assays [14,28]. However, in a recent manuscript by Lopez Perez J. and colleagues (2025), the authors have defined the AFIAS system a valid method for monitoring anti-TNF-α inhibitors; however, the interchangeability with ELISA should be further confirmed [27]. Here, to the best of our knowledge, we evaluate for the first time the analytical performance of both AFIAS ADL and AFIAS IFX in serum samples collected from pediatric patients during the routine clinical practice.

Passing–Bablok regression and Bland–Altman plots were used to assess the agreement and relative differences between analytical methods, as previously described [14,26,28]. In our hands, Passing–Bablok regressions for both IFX and ADL drugs’ concentrations revealed absence of a constant bias, whereas only for ADL, the 95% CI slopes did not include 1, indicating the presence of a slight proportional bias. Moreover, the Spearman’s coefficient was close to 1 for both IFX and ADL and revealed a significant correlation between the ELISA and AFIAS assays (*p* < 0.001) (Figure 1A,B).

The Bland–Altman test showed that the % mean difference between drug concentrations measured by ELISA and AFIAS was −14.09 for IFX (Figure 1C) and 15.79 for ADL (Figure 1D). According to previously reported acceptance criteria, for IFX, at least 66% of paired samples showed a % difference within ±20%, whereas for ADL, only 46% of the analyzed samples were within this acceptable range [38]. The higher deviation observed for ADL compared to IFX was also evident in the analysis of linear regression residuals (Appendix A).

However, our results are in line with previously published reports for AFIAS IFX and ADL [14,28].

It is also worthwhile to say that despite a comparable measurement range for IFX and ADL drug levels (0.20–50 µg/mL for AFIAS vs. 0.40–45.0 µg/mL for ELISA), the higher % differences observed for adalimumab drug levels could be partially explained by an overestimation of AFIAS results compared to ELISA measurements. This observation confirms previous published data and could explain the higher % differences observed for adalimumab rather than for infliximab [28]. However, when comparing only ELISA and AFIAS paired concentrations falling within the suggested therapeutic range (5–10 µg/mL), at least 60% of paired samples showed a % difference within the acceptable range of ±20% [38].

Similarly, we qualitatively evaluated the inter-rater agreement with kappa by stratifying both ELISA and AFIAS drug results into three subgroups according to the IFX and ADL therapeutic interval. In particular, for IFX, we divided measurements as follows (<3 µg/mL, ≥3 to ≤7 µg/mL, and ≥7 µg/mL), based on a previously described therapeutic range of 5–7 µg/mL [26,29]. Infliximab levels showed “substantial agreement” between the two assays according to the scale suggested by Landis and Koch, G.G. [31]. In particular, the weighted kappa was 0.73 (95% CI 0.58 to 0.87), and the number of observed agreements between the reference ELISA and the AFIAS assay was 33/44 (75% of the observations). Similarly, ADL drug levels were stratified into three subgroups (<5 µg/mL, ≥5 to ≤10 µg/mL, and ≥10 µg/mL) according to the therapeutic range of 5–10 µg/mL proposed for inflammatory bowel diseases [30]. For ADL, the weighted kappa was 0.52 (95% CI 0.34 to 0.71), denoting a “moderate agreement” compared to IFX [31]. In particular, the overall number of observed agreements between reference ELISA and AFIAS assay was 35/54 (64.8% of the observations) for ADL. Similar to the observations made for the Bland–Altman results, the reasons behind this “moderate agreement” for adalimumab drug levels could be probably due to the overestimation observed for the AFIAS results compared to ELISA concentrations. In fact, the proportional bias observed between ELISA and AFIAS ADL measurements has been previously described and could additionally explain the discrepancy shown for AFIAS ADL compared to IFX [27].

However, it is worth mentioning that for therapies with biologic agents, slight variations in drug measurement are unlikely to significantly affect the clinical decision process, since dosing is usually guided by therapeutic ranges that are tolerant to small variations in the drug levels. Therefore, we believe that, in our study, this discrepancy does not represent a strong limitation in clinical decision making.

Finally, alongside infliximab serum levels, we measured the amount of total anti-IFX antibodies. The most common cause of failure in biological therapy is the development of anti-drug antibodies [39]. The presence of ADAs vs. IFX is generally associated with a reduced serum IFX concentration, decreased clinical response to IFX, and increased adverse events [40]. Moreover, the presence of anti-ADL antibodies was associated with low or undetectable serum trough ADL levels and reduced clinical efficacy [41]. This aspect was further confirmed by empirical evidence showing that the concomitant use of immunosuppressive drugs reduces immunogenicity and overall ADAs production [39]. Therefore, monitoring ADAs alongside drug levels allows for the identification of patients who have lost response to the biological agent but also to potentially define a tailored therapeutic strategy [16]. Moreover, monitoring antibodies against IFX in CD patients is more cost-effective than dose-escalating strategies [42,43]. Based on this evidence, several bioanalytical methods have been proposed for the detection of antidrug antibodies, including ELISA, RIA, ECLIA, etc. [39,44,45]. POC devices have also been validated by comparing their clinical performances vs. ELISA assays [28,34,36,37]. Despite the availability of different bioanalytical methods, some questions remain to be answered. Although several TDM-based algorithms have been proposed in real-life clinical practice to guide the tailored optimization of anti-TNF therapies [15,23,46], reference ranges for anti-drug antibodies are still lacking. Moreover, according to different analytical performances, the detection of anti-drug antibodies is often based on cut-off values that provide qualitative or semi-quantitative results. Consequently, the interpretation of data is not straightforward for clinicians. Similarly, from a clinical point of view, the need for routine measurement of anti-drug antibodies alongside drug levels, or just following low-drug C*_trough_* detection, is still debated. Finally, the clinical evaluation of free rather than total anti-drug antibodies remains an open question. Together, these aspects represent a significant limitation that hampers both the measurement and subsequent interpretation of ADA levels in routine clinical practice.

Here, we compared the semi-quantitative performance of both ELISA and AFIAS assays. However, because the AFIAS Total Anti-IFX provides a working range of 8–250 AU/mL, values below or above this range are exclusively displayed as <8 and >250 AU/mL, respectively. In contrast, the analysis of total anti-IFX serum levels by ELISA can provide numeric results for antibody concentrations <8 or >250 AU/mL. Therefore, we were not able to pair sufficient data to establish a statistically significant correlation between AFIAS and ELISA results. However, n = 39 serum samples were analyzed using AFIAS, and the results were <8 AU/mL. These samples were also tested using ELISA, resulting in a median (range) value of 4.78 (2.22–12.31) AU/mL. Similarly, n = 4 serum samples analyzed with AFIAS had a total anti-IFX value of >250 AU/mL. The median value obtained for these samples following ELISA assessment was 576.5 AU/mL (range 363.1–771.8 AU/mL). Consequently, we can only assume a positive correlation between anti-IFX levels measured using both ELISA and AFIAS assays. This aspect represents a limitation of our study. Perhaps a larger number of samples could be useful to overcome the variability observed for ADL drug concentrations. Similarly, by including a higher number of samples, it could be possible to provide a more robust correlation between ADA levels measured with both methods. However, it is also worth noting that from a clinical point of view, and in the absence of reference ranges for anti-drug levels, the utility of providing a numeric value whenever the ADAs concentration is <8 or >250 AU/mL remains debated. Another limitation of our study could be the fact that the ELISA and AFIAS assays were not performed at the same time. In fact, whereas the AFIAS tests have been executed on fresh serum samples, ELISA was carried out after one freezing–thawing cycle. To eliminate this potential bias among methods, four authentic samples of IFX and ADL were randomly selected and re-analyzed after being stored at −80 °C for one month. Compared to the initial drugs’ concentrations, a second cycle of freezing–thawing showed a 30-day stability of 119% and 97.52% for IFX and ADL, respectively. Therefore, these data demonstrate that ELISA results are not affected by the freezing–thawing procedures.

## 5. Conclusions

In conclusion, our results show that, although there are slight differences between infliximab and adalimumab, the AFIAS assay has an accuracy and analytical performance comparable to that of the ELISA method used for TDM of IFX and ADL drug levels. Regarding the evaluation of anti-IFX antibodies, we only provided a preliminary comparison between ELISA and AFIAS tests, highlighting the necessity for more exhaustive studies aimed at addressing this point. However, our data seem to show a positive correlation between total anti-IFX levels measured using both bioanalytical methods. AFIAS IFX and ADL have been already validated as lateral flow immunoassays for the quantitative measurement of infliximab and adalimumab in different biological matrices. As a point-of-care device, the AFIAS is based on a rapid and fully automated process that does not require specialized or well-trained personnel and allows single-sample analyses with results displayed within 15–20 min. The latest represents the main advantage of this method, which could overcome the time- and cost-consuming limitations of current ELISA-based TDM approaches. Therefore, based on our results, the introduction of these devices into routine clinical practice could provide results more quickly and with similar accuracy to ELISA, allowing clinicians to rapidly formulate effective therapeutic plans. In fact, an ad hoc protocol has been established in our center to promote the use of this device, encouraging a proactive TDM strategy and, therefore, facilitating a more cost-effective strategy than the empirical dose-escalating approach.

## Figures and Tables

**Figure 1 pharmaceutics-17-00421-f001:**
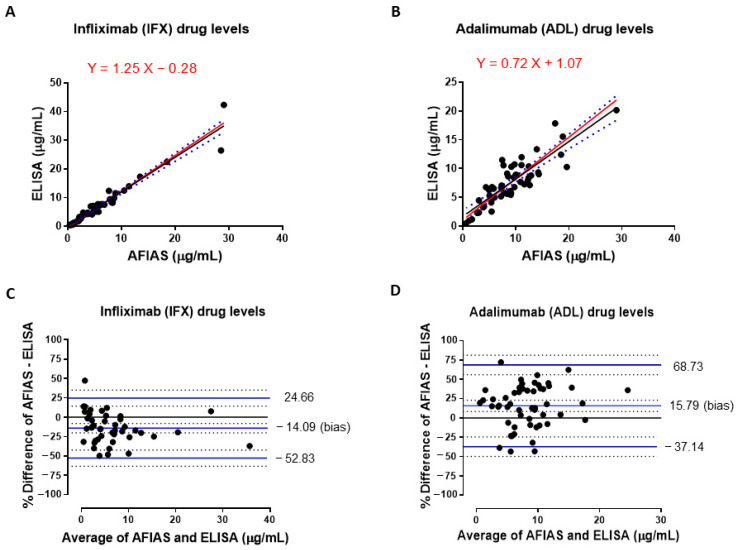
Passing–Bablok correlations and Bland–Altman plots for infliximab (IFX) and adalimumab (ADL) drug levels. Passing–Bablok correlation plots for IFX (**A**) and ADL (**B**) drug levels. Continuous black and red lines indicate the linear and the Deming regression lines, respectively. The dotted blue lines indicate 95% confidence intervals (CIs). Bland–Altman plots of IFX (**C**) and ADL (**D**) drug levels. Continuous horizontal blue lines were drawn from top to down at the upper 95% limits of agreement, the mean difference (bias), and the lower 95% limits of agreement. Dotted horizontal black lines indicate the 95% confidence interval (CI) for the mean and the agreement limits (defined as the mean difference ± 1.96 times the standard deviation of the differences).

**Table 1 pharmaceutics-17-00421-t001:** Comparison between infliximab serum levels measured by enzyme-linked immunosorbent assay (ELISA) and AFIAS stratified by the therapeutic range 3–7 µg/mL [29].

Reference ELISA			
(IDKmonitor^®^ Infliximab Drug Levels)			
	<3 µg/mL	≥3 to <7 µg/mL	≥7 µg/mL
AFIAS Infliximab			
<3 µg/mL	13	5	0
≥3 to <7 µg/mL	0	8	6
≥7 µg/mL	0	0	12

**Table 2 pharmaceutics-17-00421-t002:** Comparison between adalimumab serum levels measured by enzyme-linked immunosorbent assay (ELISA) and AFIAS stratified by the therapeutic range 5–10 µg/mL [30].

Reference ELISA			
(IDKmonitor^®^ Adalimumab Drug Levels)			
	<5 µg/mL	≥5 to <10 µg/mL	≥10 µg/mL
AFIAS Adalimumab			
<5 µg/mL	8	2	0
≥5 to <10 µg/mL	4	18	4
≥10 µg/mL	0	9	9

## Data Availability

The original contributions presented in this study are included in the article/Appendix A. Further inquiries can be directed to the corresponding authors.

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
