# Peer review of "Measurement of Anti-TNF Biologics in Serum Samples of Pediatric Patients: Comparison of Enzyme-Linked Immunosorbent Assay (ELISA) with a Rapid and Automated Fluorescence-Based Lateral Flow Immunoassay"

_pharmaceutics, 2025, doi:10.3390/pharmaceutics17040421_

Round 1
Reviewer 1 Report (Previous Reviewer 3)
Comments and Suggestions for Authors
The manuscript submitted by Chiara Rossi et al. describes the comparison of analytical performance of two assays for the measurements of two anti-TNF drugs: clinically validated ELISA (Immundiagnostik AG) and a novel fluorescence-based lateral flow assay AFIAS ( Boditech Med Inc.). As authors indicate similar comparative studies have been conducted recently and publishes in 2024-2025 (references 14,27,28 in the manuscript) for Therapeutic Drug Monitoring of Infliximab and Adalimumab. However, the study is independent and conducted obviously in same period of time as the already published and referred in the manuscript studies. Moreover, the samples are collected from paediatric patients and this detail may be of special interest. My previous remarks to the authors have been answered and thus I recommend the acceptance of the manuscript for publication.
Author Response
The manuscript submitted by Chiara Rossi et al. describes the comparison of analytical performance of two assays for the measurements of two anti-TNF drugs: clinically validated ELISA (Immundiagnostik AG) and a novel fluorescence-based lateral flow assay AFIAS ( Boditech Med Inc.). As authors indicate similar comparative studies have been conducted recently and publishes in 2024-2025 (references 14,27,28 in the manuscript) for Therapeutic Drug Monitoring of Infliximab and Adalimumab. However, the study is independent and conducted obviously in same period of time as the already published and referred in the manuscript studies. Moreover, the samples are collected from paediatric patients and this detail may be of special interest. My previous remarks to the authors have been answered and thus I recommend the acceptance of the manuscript for publication.
Many thanks to Reviewer#1 for appreciating our revision. We think that his/her comments have significantly improved our manuscript.
Reviewer 2 Report (Previous Reviewer 2)
Comments and Suggestions for Authors
The article has been improved in this round of revision and my major concerns have been addressed. However, I still consider Fig 1 as sub-par quality. All legends need to go into Fig caption and sub-charts and fonts increased.
Author Response
The article has been improved in this round of revision and my major concerns have been addressed. However, I still consider Fig 1 as sub-par quality. All legends need to go into Fig caption and sub-charts and fonts increased.
Many thanks to Reviewer #2 for appreciating our revision. Following his/her suggestion, we have improved Figure 1 resolution by increasing the dots per inch (dpi) value to 600 as indicated in the authors guidelines. Similarly, we have augmented the font within each graph and included the legend caption that was missing in the initial submission.
Reviewer 3 Report (Previous Reviewer 1)
Comments and Suggestions for Authors
The article has been modified and is now suitable for publication. I thank the authors for clarifying the points raised and adding the table to the supplementary file.
Author Response
The article has been modified and is now suitable for publication. I thank the authors for clarifying the points raised and adding the table to the supplementary file.
Many thanks to Reviewer #3 for appreciating the revised version of our manuscript. In our opinion, these comments have greatly improved the quality of our research.
This manuscript is a resubmission of an earlier submission. The following is a list of the peer review reports and author responses from that submission.
Round 1
Reviewer 1 Report
Comments and Suggestions for Authors
The manuscript compares an ELISA assay with a lateral flow assay to detect anti-TNF therapy biologics. The rationale is adequate, and the clinical is obvious. The methodology of ELISA is validated; on the other hand, there is not much information on lateral flow, not even on the company's website. It is essential to analyze the validation of the protocol. Another issue is the range, which is quite broad, and the reading by a specific apparatus. Even though the advantage is the time frame, more data on validation is needed. The report has a very low number of cases in which a statistical analysis was performed, but it is insufficient to provide a good report to validate the methodology. More samples are required, and more information on validation is needed. Three critical questions: the detection system's linearity, the method's sensitivity to hemolyzed samples, and the assay's specificity. These elements are required before reevaluating the manuscript.
Author Response
Reviewer 1
The manuscript compares an ELISA assay with a lateral flow assay to detect anti-TNF therapy biologics. The rationale is adequate, and the clinical is obvious. The methodology of ELISA is validated; on the other hand, there is not much information on lateral flow, not even on the company's website. It is essential to analyze the validation of the protocol. Another issue is the range, which is quite broad, and the reading by a specific apparatus. Even though the advantage is the time frame, more data on validation is needed. The report has a very low number of cases in which a statistical analysis was performed, but it is insufficient to provide a good report to validate the methodology. More samples are required, and more information on validation is needed. Three critical questions: the detection system's linearity, the method's sensitivity to hemolyzed samples, and the assay's specificity. These elements are required before reevaluating the manuscript.
Response:
Many thanks to Reviewer #1 for highlighting this interesting point. As described in Materials & Methods section, infliximab (IFX) and adalimumab (ADL) drug levels were measured by using a fluorescence-based lateral flow immunoassay developed and validated by Bodytech Med Inc., (South Korea). Similarly, total anti-IFX antibodies have been tested by using the same strategy. These assays have been designed for in vitro diagnostic use and allow the quantitative (for IFX and ADL drug levels) and semi-quantitative (for anti-drug antibodies) analysis in whole blood, serum and plasma. Therefore, the bioanalytical validation has been already conducted before being commercialized. Actually, kits used in our study are available on market with the following catalogue numbers (Cat. n.): SMFP-75 for AFIAS infliximab, SMFP-89 AFIAS for adalimumab and SMFP-76 AFIAS for Total Anti-Infliximab (Bodytech Med Inc., South Korea). Details on the bioanalytical validation have been reported in the technical datasheets and are available on request at TS@boditech.co.kr. Additionally, data on the development and the analytical performance of this technology have been already published in different clinical settings:
- Kim, E.S.; Chon, H.; Kwon, Y.; Lee, M.; Kim, M.J.; Choe, Y.H. Fluorescence-Based Lateral Flow Immunoassay for Quantification of Infliximab: Analytical and Clinical Performance Evaluation. Ther Drug Monit 2024, 46, 460-467, doi:10.1097/FTD.0000000000001176.
-Iniesta-Navalón, C.; Ríos-Saorín, M.; Añez-Castaño, R.; Rentero-Redondo, L.; Ortíz-Fernandez, P.; Martínez, E.; Urbieta-Sanz, E. Evaluating the Accuracy and Clinical Utility of AFIAS-10 Point of Care Versus Enzyme-Linked Immunosorbent Assay in Therapeutic Drug Monitoring of Infliximab and Adalimumab. Therapeutic Drug Monitoring 2024, doi:10.1097/FTD.0000000000001269.
- Lopez Perez, J.; Inda-Landaluce, M.; Nocito-Colon, M.; Martinez-Lostao, L. Comparative Analysis of 2 Commercially Available Assays for Therapeutic Drug Monitoring of Infliximab and Adalimumab. Ther Drug Monit 2025, doi:10.1097/FTD.0000000000001295.
In particular, the latest two published articles have used the AFIAS-10 instead of AFIAS-6 adopting the same design described in our manuscript, and none validation data have been presented. Therefore, our aim was not to further validate the AFIAS test but to compare the analytical
performances of this new fluorescence immunoassay with a reference ELISA, in order to provide a valid and quicker alternative to the gold standard method for the measurement of IFX, ADL and anti-IFX antibodies in pediatric patients during the routine clinical practice. In fact, by using a more rapid diagnostic tool we aim to promote among clinicians a proactive TDM strategy instead of recurring to a reactive approach that sometimes is also characterized by an unfavorable cost-benefits ratio.
Reviewer 2 Report
Comments and Suggestions for Authors
The manuscript “Measurement of anti-TNF biologics in serum samples of pediatric patients: comparison of enzyme-linked immunosorbent assay (ELISA) with a rapid and automated fluorescence-based lateral flow immunoassay” by Rossi et al. is devoted to the validation of fluorescence-based lateral flow immunoassay (AFIAS) for therapeutic drug monitoring. They compared it against the gold standard (ELISA).
It is a well-planned and well-written article. It can be of interest to broad scientific and clinical audiences.
Shortcomings:
1. Novelty needs to be stated in the introduction. It is subtly stated in lines 418-419
2. Limitations need to be discussed.
3. Figure 1 is not journal quality.
4. Repetitions in Introduction and Discussion. For example, lines 120-127 and 379-390 are very similar.
5. Lines 401-413 are not necessary in the Discussion. It was described previously.
Author Response
Reviewer 2:
The manuscript “Measurement of anti-TNF biologics in serum samples of pediatric patients: comparison of enzyme-linked immunosorbent assay (ELISA) with a rapid and automated fluorescence-based lateral flow immunoassay” by Rossi et al. is devoted to the validation of fluorescence-based lateral flow immunoassay (AFIAS) for therapeutic drug monitoring. They compared it against the gold standard (ELISA).
It is a well-planned and well-written article. It can be of interest to broad scientific and clinical audiences.
Response:Many thanks to Reviewer #2 for appreciating our manuscript.
Shortcomings:
- Novelty needs to be stated in the introduction. It is subtly stated in lines 418-419
Response:1. Many thanks for this important suggestion. Novelty of our study has been now stated in the Introduction.
- Limitations need to be discussed.
Response:2. We do apologies for this missing point. We have now reported the main limitations of our study in the Discussion.
- Figure 1 is not journal quality.
Response:3. Thanks to Reviewer #2 for pointing out this issue. Resolution of Figure 1 has been now increased to 600 dpi as reported in the instructions for authors.
- Repetitions in Introduction and Discussion. For example, lines 120-127 and 379-390 are very similar.
Response:4. We do apologies for these repetitions that have been now removed from the Introduction and Discussion sections.
- Lines 401-413 are not necessary in the Discussion. It was described previously.
Response:5. May thanks to Reviewer #2 for this suggestion. The highlighted lines have been now removed from the Discussion section.
Reviewer 3 Report
Comments and Suggestions for Authors
The manuscript submitted by Chiara Rossi et al. describes the validation of a new immunoassay for TDM of two IBD drugs by comparing its results with those of a clinically validated ELISA method. The authors demonstrate that the two methods have substantial agreement for Infliximab with Cohen's kappa 0,73 and moderate for adalimumab with Cohen's kappa 0,53. There is a positive correlation between the two methods' measurements, however, proportional bias was detected for adalimumab. The manuscript is well-structured, methods are accurately described and results are clearly presented.
But I have some remarks that should be addressed/answered before manuscript acceptance for publication.
1. I think that the concentrations found by two methods (raw data) should be presented in the supplements (supplementary table).
2. The samples were not equally treated. The samples for the ELISA method were frozen. Are the authors sure that the freezing-thawing procedure does not introduce some bias between the two methods?
3. The authors used Passing-Bablok regression for method comparison. Did the authors apply the CUSUM test for this regression validity?
4. Lines 184-186 of the manuscript - possibly mistakes in concentration units, not ng/mL by μg/mL.
5. Line 285 "Moreover, the 95% CI did not include 1, indicating the presence of significant correlation (p < 0.001)." How 95% CI can indicate the significance of correlation?
Author Response
Reviewer 3:
The manuscript submitted by Chiara Rossi et al. describes the validation of a new immunoassay for TDM of two IBD drugs by comparing its results with those of a clinically validated ELISA method. The authors demonstrate that the two methods have substantial agreement for Infliximab with Cohen's kappa 0,73 and moderate for adalimumab with Cohen's kappa 0,53. There is a positive correlation between the two methods' measurements, however, proportional bias was detected for adalimumab. The manuscript is well-structured, methods are accurately described and results are clearly presented.
But I have some remarks that should be addressed/answered before manuscript acceptance for publication.
- I think that the concentrations found by two methods (raw data) should be presented in the supplements (supplementary table).
Response:1. Many thanks to Reviewer #3 for this suggestion. We have now included a Supplementary Table 3, in which raw data for both infliximab and adalimumab concentrations are displayed.
- The samples were not equally treated. The samples for the ELISA method were frozen. Are the authors sure that the freezing-thawing procedure does not introduce some bias between the two methods?
Response:2. We thank Reviewer #3 for raising this question. We are well aware that serum sample were frozen before ELISA determination and that a source of variability could be introduced between two methods. However, in order to eliminate a potential bias, n=4 authentic samples of IFX and ADL were randomly selected and re-analyzed after being stored at -80 °C for one month. Compared to the initial drugs’ concentrations, a second cycle of freezing-thawing showed a 30-days stability of 119 % and 97.52 % for IFX and ADL, respectively. Therefore, these data demonstrate that ELISA results are not affected by the freezing-thawing procedure.
- The authors used Passing-Bablok regression for method comparison. Did the authors apply the CUSUM test for this regression validity?
Response:3. We’d like to thank Reviewer #3 for highlighting this point. Integrity of linear regression has been evaluated by plotting residuals vs drug concentrations measured with AFIAS test. Graphs of residuals for both infliximab and adalimumab are now displayed in Supplementary Figure 1.
- Lines 184-186 of the manuscript - possibly mistakes in concentration units, not ng/mL by μg/mL.
Response:4. Thanks for raising this observation. The concentration units (ng/mL) reported in the Materials and Methods section relative to the ELISA assay are correct since both IDKmonitor® infliximab and adalimumab drug levels kits present a calibration curve ranging from 0.00 to 225.00 ng/mL. However, concentration results are expressed in μg/mL since a dilution factor of 200 must be applied to get the final drug concentrations for each sample. However, a sentence in the ELISA paragraph has been included to clarify this point.
- Line 285 "Moreover, the 95% CI did not include 1, indicating the presence of significant correlation (p < 0.001)." How 95% CI can indicate the significance of correlation?
Response:5. We do apologies for this typesetting mistake that has been now modified in this revised manuscript.